# How to Improve TRUS-Guided Target Biopsy following Prostate MRI

**DOI:** 10.3390/cancers13225647

**Published:** 2021-11-11

**Authors:** Byung Kwan Park

**Affiliations:** Department of Radiology, Samsung Medical Center, Sungkyunkwan University School of Medicine, Seoul 06351, Korea; rapark@skku.edu; Tel.: +82-2-3410-6457

**Keywords:** prostate cancer, transrectal ultrasound, biopsy, magnetic resonance imaging

## Abstract

**Simple Summary:**

Radiologists or urologists prefer to use transrectal ultrasound (TRUS) for detecting a prostate cancer. Therefore, it is of great importance to depict and target an index lesion with TRUS after the prostate MRI is scanned. They need to know the new TRUS protocols, imaging features, and biopsy techniques. The new TRUS protocols include using fundamental imaging rather than harmonic imaging and lowering dynamic range to increase tumor-to-normal tissue contrast. The new TRUS features show how to identify an index lesion and how to differentiate insignificant and significant cancers in terms of tumor size, echogenicity, echotexture, margin, and perfusion. The new biopsy techniques include how to understand different tumor locations, sizes, and shapes between magnetic resonance imaging (MRI) and TRUS and how to target an index lesion regarding biopsy strategy and cores. Systematic biopsy is necessary but can be skipped in patients with invasive behaviors such as extra-capsular extension, seminal vesicle invasion, or metastasis. Image fusion biopsy as well as cognitive biopsy can be improved if radiologists or urologists are familiar with the new TRUS protocol, imaging features, and biopsy techniques.

**Abstract:**

TRUS is a basic imaging modality when radiologists or urologists perform cognitive fusion or image fusion biopsy. This modality plays the role of the background images to add to an operator’s cognitive function or MRI images. Operators need to know how to make TRUS protocols for lesion detection or targeting. Tumor location, size, and shape on TRUS are different from those on MRI because the scan axis is different. TRUS findings of peripheral or transition tumors are not well known to radiologists and urologists. Moreover, it remains unclear if systematic biopsy is necessary after a tumor is targeted. The purpose of this review is to introduce new TRUS protocols, new imaging features, new biopsy techniques, and to assess the necessity of systematic biopsy for improving biopsy outcomes.

## 1. Introduction

Recently, we have experienced a transition period in diagnosing prostate cancer. Digital rectal examination has been used for clinical staging instead of MRI examination [1,2,3]. However, most cancers that develop in the anterior compartment of the prostate are barely palpable [4,5,6]. The software and hardware of MRI scanners have been developed to improve MR image quality and PI-RADS has been introduced and used in clinical practice [7,8,9,10]. The number of prostate MRI examinations has been increasing prior to biopsy because PI-RADS is known to be useful for stratifying the risk of prostate cancers [11,12,13,14]. Many studies have reported that pre-biopsy MRI contributes to detecting significant cancers. A PI-RADS score of four or five is a strong indication for a biopsy due to higher likelihood of significant cancer compared to a PI-RADS score of 1–3 [7,9].

Currently, radiologists and urologists use TRUS to perform a cognitive fusion or image fusion biopsy following an MRI scan [15,16,17,18,19,20,21,22,23]. However, they often do not know how to make TRUS protocols, how to read imaging findings, and how to perform biopsy techniques for lesion detection and targeting [24,25,26,27,28,29]. TRUS scans are of great importance for taking a precise biopsy when cognitive fusion or imaging fusion is performed. A good quality of TRUS image is essential to depict or target an index tumor, which is detected on MRI. Lesion location or imaging feature on TRUS are quite different from those on MRI. Moreover, it remains unclear whether a systematic biopsy is necessary. Theoretically, as the PI-RADS score increases, the likelihood of significant cancer becomes higher, suggesting that a target biopsy would be sufficient to detect significant cancer. However, several reports indicate that systematic biopsy contributes to detecting additional significant cancers [24,25,26,28,29].

The purpose of this review is to introduce new TRUS protocols, new imaging features, new biopsy techniques, and to assess the necessity of systematic biopsy for improving biopsy outcomes.

## 2. TRUS Imaging Protocols

Currently, TRUS scanners are designed to prefer harmonic imaging rather than fundamental imaging because of better axial and lateral resolutions (Figure 1) [30,31,32]. Many types of artifacts frequently develop on fundamental imaging as the US travels through various organs with different tissue densities (Table 1) [30]. Harmonic imaging can reduce these US artifacts which do not contribute to making good quality TRUS imaging. Non-linear US propagation through tissues is exploited for harmonic imaging, which takes advantage of the fact that high-pressure US waves go faster than low-pressure waves [30,32]. It results in distorting the shape of the US wave. Accordingly, harmonics are generated from this waveform change in a tissue. Harmonic waves increase to some depth and thereafter decrease because of attenuation. However, increasing tissue resolution decreases tissue contrast between prostate cancer and normal tissue. When a tumor is located near the transducer, the US contrast between tumor and normal tissue may decrease due to the lack of harmonics. In contrast, fundamental imaging provides higher tissue contrast compared to harmonic imaging despite its lower image resolution (Figure 1).

The TRUS dynamic range should be kept less than 50 to enhance the tissue contrast by sharpening a tumor edge [33]. Thus, a low dynamic range combined with fundamental imaging can maximize the tumor-to-normal tissue contrast (Figure 2) [25,26,29]. As the dynamic range decreases, tissue contrast increases, but image resolution decreases (Figure 2). Image quality will become very poor if the dynamic range is too low. Therefore, radiologists or urologists should control the image quality of TRUS not only to maximize the tumor-to-normal tissue contrast, but also to minimize loss of image resolution. Radiologists and urologists should keep in mind that the optimal dynamic ranges differ somewhat among commercially available US scanners. They need to investigate which dynamic ranges are optimal for tissue contrast to discriminate prostate cancer from normal tissue without significantly sacrificing image resolution.

The quality of a TRUS image is influenced mainly by two key protocols such as using fundamental imaging and low dynamic range to increase tissue contrast between index tumor and normal tissue. If radiologists or urologists become familiar with these TRUS protocols, they can produce reproducible TRUS images in which an index lesion is identified.

## 3. TRUS Imaging Features

Traditionally, radiologists and urologists have believed that prostate cancer is hypoechoic on TRUS (Figure 2 and Figure 3) [34,35]. However, this imaging finding is consistent only with peripheral cancer (Table 2) [29]. Moreover, many false positive lesions can mimic hypoechoic peripheral cancers, such as inflammation, infarction, and peripheral hyperplastic nodules [36]. Color Doppler TRUS can add value in detecting peripheral cancer, in which blood flow is increased due to increased vascularity (Figure 3) [36]. PI-RADS v2.0 and 2.1 also accept the utility of lesion enhancement in upgrading a PI-RADS 3 into 4. Therefore, color Doppler TRUS can be used to improve the detection of a peripheral cancer seen on MRI.

Previous studies have shown that hyperechoic cancers might account for 40% of cases [36,37,38,39]. However, they did not demonstrate that these hyperechoic tumors came from the transition zone. Recent papers have reported that transition cancer is not hypoechoic, but hyperechoic compared to neighboring transitional tissue (Table 2) (Figure 4 and Figure 5) [26,27,29]. Chung et al. reported that a hypoechoic rim around a tumor is another useful finding in detecting a transition cancer (Figure 5) [27]. This TRUS finding is rare in a BPH nodule or a peripheral cancer. Color Doppler TRUS is not useful in detecting a transition cancer for the same reason that PI-RADS v2.1 does not use DCEI in scoring transition lesions. We need to precisely correlate MRI and TRUS findings to determine whether a focal lesion on TRUS is the same as that on MRI. Lesion-by-lesion correlation between MRI and TRUS is of great importance, not only to precisely target a tumor with TRUS, but also to exclude false positive lesions.

TRUS as well as MRI can suggest imaging findings, suggestive of significant cancer (Table 2). As a PI-RADS score increases, a tumor’s signal intensity decreases on T2WI or increases on DWI [7,8,9,10]. Consequently, the tumor detection improves on MRI. Similarly, a PI-RADS 5 lesion appears more hypoechoic for significant peripheral cancer and more hyperechoic for significant transition cancer on TRUS (Figure 2, Figure 3, Figure 4, Figure 5, Figure 6, Figure 7 and Figure 8) [25,26,27,29]. Thus, the tumor conspicuity becomes clearer on TRUS as the PI-RADS score increases. Tumor detection is easier on TRUS when a prostate cancer becomes significant because GS increases (Figure 2, Figure 3, Figure 4, Figure 5, Figure 6, Figure 7 and Figure 8) [40,41,42].

Assessing the tumor margin also can offer useful information on significant cancer (Table 2). If it is not smooth but irregular or spiculate, the tumor can suggest significant cancer rather than insignificant cancer (Figure 2, Figure 3, Figure 4, Figure 5, Figure 6, Figure 7 and Figure 8) [27]. As a prostate cancer becomes significant, tumor margin becomes irregular, spiculate, or infiltrative. PI-RADS version 2.1 does not consider the tumor margin because it is not easy to assess on MRI. However, TRUS can describe the tumor margin to determine whether it is smooth. Moreover, if the presence of ECE is inconclusive on MRI, TRUS can help to determine whether it is present (Figure 3 and Figure 7).

Other TRUS features suggesting significant cancers include increasing tumor size and heterogeneous echotexture (Table 2) (Figure 2, Figure 3, Figure 4, Figure 5, Figure 6, Figure 7 and Figure 8) [27,43]. The significant cancer detection rate in a small PI-RADS 4 lesion, which is less than 1 cm, is lower than that in a larger PI-RADS 4 lesion [43]. The tumor size is also an independent factor for differentiating PI-RADS 4 and 5 lesions. As a PI-RADS score increases, the tumor may have higher GSs, which makes their echotexture heterogeneous because of gland fusion, absent stroma, absent glands, and tumor necrosis [44]. Therefore, tumors with increasing GSs are supposed to become heterogeneous in echotexture (Figure 2, Figure 3, Figure 4, Figure 5, Figure 6, Figure 7 and Figure 8).

## 4. TRUS Biopsy Techniques

Radiologists and urologists need to be familiar with the following biopsy techniques for lesion detection on TRUS (Table 3). First, prostate compression should be minimized to improve tumor detection during TRUS (Figure 2, Figure 4, Figure 6, Figure 7 and Figure 8) [24,25,26,28,29]. When a transducer is introduced into the rectum, the prostate is likely to be deformed into a banana shape (Figure 3 and Figure 5). Accordingly, the shape of a small prostate cancer can become so changed that radiologists or urologists cannot easily detect it because they are expecting the tumor shape seen on the MRI. Additionally, it is not uncommon for a small cancer to be embedded into the parenchyma when it is located in the peripheral zone around the Denonvillier fascia. Therefore, minimizing prostate compression is required to improve lesion detection. Biopsy beginners are likely to compress the prostate, which makes it difficult to detect a prostate cancer [45].

Second, a lesion on TRUS appears more superiorly than that on MRI as it becomes closer to the posterior capsule (Figure 8) [24,25,26,28,29]. However, the tumor on TRUS appears more inferiorly than that on MRI as it becomes closer to the anterior capsule. This phenomenon results from the different scan axes of TRUS and MRI (Figure 9) [45]. Generally, the scan axis of MRI is perpendicular to urethra, whereas that of TRUS is oblique to urethra. Therefore, as a lesion becomes closer to posterior capsule, it is located higher on TRUS compared to MRI. When MRI shows a tumor at the apex, TRUS shows it at the mid-gland or base. In contrast, if a lesion becomes closer to anterior capsule, it is seen lower on TRUS (Figure 4 and Figure 5). When MRI shows a tumor at the base, TRUS shows it at the mid-gland or apex. A tumor which is located around the transverse line between anterior and posterior capsules, the location is not so different between MRI and TRUS (Figure 7). Additionally, patient positions such as supine, decubitus, or knee-chest position influence differences in the apparent lesion locations between MRI and TRUS.

Third, the shape or size of a lesion on TRUS is different from those on MRI due to the different scan axes (Figure 4, Figure 5, Figure 7 and Figure 8) [24,25,26,28,29]. When an oval tumor is shown on MRI, it can be round on TRUS. The tumor size measured on MRI can become longer or shorter on TRUS. Radiologists and urologists might miss a lesion on TRUS if they do not expect a discrepancy between the MRI and TRUS results in terms of imaging features. Therefore, when looking for a tumor on TRUS, radiologists or urologists should keep in mind that the lesion’s location, size, and shape can differ from those on MRI. Otherwise, it is difficult to detect and target a lesion with TRUS when they perform a TRUS-guided biopsy.

Minimizing prostate compression with the transrectal transducer and understanding discrepant imaging features are key techniques for improving tumor targeting with TRUS [26]. Currently, MRI-TRUS image fusion biopsies are widely performed in many institutes, especially by urologists who are not used to MRI interpretation or PI-RADS scoring, and thus prefer to use image fusion biopsy rather than cognitive fusion biopsy. They encounter discrepant imaging features between MRI and TRUS because they are unfamiliar with the new biopsy techniques as mentioned above. If they minimize prostate compression and understand the mechanism of discrepant imaging features, their image fusion become more precise, which will improve their biopsy results [26].

## 5. Target Biopsy Techniques

Generally, radiologists and urologists try to target the center of the tumor during a prostate biopsy (Table 4). They believe that core tissue samples from the tumor center represent the highest GS. Frequently, they obtain one or two cores from the center of a tumor. However, as the GS becomes higher, the prostate cancer becomes heterogeneous in histologic texture (Figure 2 and Figure 10) [44]. Therefore, a saturation target biopsy is reported to be useful in reducing GS underestimation [11,46,47,48]. Sampling the central area alone may miss a higher GS in the peripheral area, leading to underestimating the risk of prostate cancer (Figure 2 and Figure 10). Therefore, tissue sampling should be obtained from the peripheral area as well as the central area of the tumor. This strategy helps to reduce GS underestimation compared to that of targeting a center of the tumor alone.

Another issue with a target biopsy is deciding how many cores are obtained from the central and peripheral areas. Theoretically, increasing the number of target cores improves the chances of finding the highest GS in the tumor, but it also increases the risk of complications such as discomfort, pain, bleeding, and acute prostatitis [28,49]. Several studies have considered the number required for a saturation biopsy during tumor detection. Reportedly, 3–4 target cores are superior to 1–2 target cores, and more than four target cores are superior to four or fewer target cores in detecting significant cancer [48]. Therefore, five or more target cores are recommended for detecting significant PCa [46,50,51]. My institute adopts the following strategy of target biopsy and the number of target cores. The first, second, and third target cores are sampled from the center of an index tumor, and the fourth, fifth, and sixth target cores are sampled from its peripheral area. Central sampling is thus followed by peripheral sampling during the target biopsy (Figure 2).

Multifocal tumor targeting provides another advantage in detecting significant cancer. The conspicuity of the tumor becomes clearer as the number of target cores increases (Figure 6). So, the second half of the target cores may have a higher GS or longer cancer length compared to the first half of the target cores (Figure 2 and Figure 6). Multifocal sampling may overestimate GS compared with prostatectomy. If biopsy cores are sampled from the area consisting of a GS 4 component, the GS will be 8 (4 + 4), which is higher than 7 (3 + 4) found in prostatectomy (Figure 10). However, overestimating GS with multifocal tumor targeting does not cause harm because it does not lead to overtreatment in patients with prostate cancer. The goal of prostate biopsy is to sample a GS 4 or higher component, and multifocal tumor targeting can contribute to meeting that aim.

Radiologists or urologists try to fire an automated gun toward the center of an index tumor. Frequently, the biopsy needle does not lie perfectly on the guideline for targeting. Therefore, biopsy cores can be obtained from the slight off-center (Figure 11). Multifocal sampling can overcome for that limitation. Tumor location and shape influence the precision of targeting. If a thin and lenticular PI-RADS 4 or 5 is abutting at the anterior capsule, it is difficult to obtain cancer tissues because a 5 mm tip of a biopsy needle cannot contain core tissue (Figure 5). Therefore, the needle needs to penetrate the anterior capsule by more than 5 mm to sample pathologic tissue [28]. If anterior capsule is penetrated, unwanted bleeding occurs. Post-biopsy compression using a transducer helps to stop the bleeding [52].

When two or more lesions with PI-RADS 4 or 5 exist, radiologists and urologists can face the question of how many lesions should be targeted. Specifically, if a PI-RADS 5 transition lesion and a PI-RADS 4 peripheral lesion are detected on both MRI and TRUS, both of these tumors should be targeted. Transition cancers tend to have relatively lower GS compared to peripheral cancers [27,53,54,55]. It is not uncommon for a higher GS adenocarcinoma to be detected in a PI-RADS 4 peripheral lesion rather than in a PI-RADS 5 transition lesion (Figure 12). In my institute, target biopsies are performed in up to three PI-RADS 4 or 5 lesions, but further investigation is necessary to determine how many PI-RADS 4 or 5 lesions should be biopsied to reduce GS underestimation.

## 6. Necessity of Systematic Biopsy

Only a few studies have addressed the utility of systematic biopsy (Table 4) [24,25,26,28,29,56]. The role of systematic biopsy in detecting significant cancer thus remains unclear. As a PI-RADS score becomes higher, the likelihood of significant cancer increases [9,57]. Therefore, only precise targeting of an index lesion with PI-RADS 4 or 5 is sufficient for obtaining the high GS. However, the likelihood of significant cancer is not 0% even when a PI-RADS 1 lesion alone is found in a patient with high PSA. Moreover, the likelihood of significant cancer is not 100% even when a PI-RADS 5 lesion alone is found in a patient with high PSA. PI-RADS 3 or 4 lesions do not have a higher likelihood of significant cancer than PI-RADS 5. Therefore, the need for a systematic biopsy exists in patients with PI-RADS scores of 1–4. Skipping a systematic biopsy is difficult in patients with a PI-RADS 1–3 lesion because a target biopsy alone does not yield a high significant cancer detection rate. It is not uncommon for a systematic biopsy to find a higher GS than a target biopsy in PI-RADS 4 or 5 lesions [24,25,26,28,29]. Significant cancer can be detected by a systematic biopsy because focal inflammation can mimic PI-RADS 3–5 lesions [58,59]. For this reason, an MRI-guided in-bore biopsy might not really be superior to an MRI-TRUS cognitive- or image-fusion biopsy in detecting significant cancer. The former biopsy technique provides a better depiction of the tumor, but it has greater difficulty in performing a systematic biopsy compared to the latter techniques. Another situation that might require a systematic biopsy is when radiologists and urologists do not have strong confidence that their tumor targeting is sufficient or that a lesion on TRUS correlates with that in the MRI. Systematic biopsies are not easy to skip in those clinical settings.

Adding a systematic biopsy increases not only the number of biopsy cores, but also the frequency of post-biopsy complications such as pain, bleeding, and infection [24,28,60]. Therefore, a systematic biopsy can be skipped in patients whose lesions show definite aggressive behaviors such as extra-capsular extension, seminal vesicle invasion, or metastasis (Figure 7) (Table 4) [26]. Additionally, the number of target cores can be reduced in these patients when anti-coagulant therapy cannot be withheld [26].

Several investigations have reported new biomarkers that can suggest disease severity in prostate cancer [61,62,63]. PSA density, 4Kscore, and risk calculators are used to determine whether prostate biopsy is necessary. These biomarkers may provide useful information because negative and positive predictive values of MRI findings are much influenced by disease prevalence. Further studies are needed about issues such as the need of a systematic biopsy at various PI-RADS scores and the optimal number of biopsy cores for systematic biopsy.

## 7. Cognitive Fusion Biopsy vs. Image Fusion Biopsy

Currently, radiologists or urologists use MRI-TRUS cognitive fusion biopsy and/or image fusion biopsy. Many studies have compared these types of TRUS-guided biopsies to determine which one is superior to the other in detecting prostate cancer [15,16,17,18,19,20,21,22,23]. However, these projects are not a well-controlled studies in terms of operator’s experience, tumor size, location, and PI-RADS score.

Urologists or radiologists who cannot interpret prostate MR images want to perform image fusion biopsies. However, image fusion is difficult to correlate lesion-by-lesion on fused images. Thus, they should repeat image fusions by using various landmarks to achieve good outcomes [15,16,17,18,19,20,21,22,23]. However, they try to fuse MRI-TRUS images even though a tumor is clearly seen on TRUS alone. Intuitively, visible and invisible tumors on TRUS will differ significantly in terms of the cancer detection rate because the former tends to produce easier and better lesion-by-lesion correlations than the latter in MRI-TRUS fusion images.

Another limitation of the MRI-TRUS image-fusion technique is the frequent failure to control for TRUS image quality and focus only on precise lesion correlation. If radiologists and urologists know optimal TRUS protocols, new imaging features, and biopsy techniques, they can detect most PI-RADS 4 or 5 lesions without relying on image fusion [26]. Therefore, if they can detect a lesion using new TRUS techniques and imaging features, it should not be defined as a cognitive biopsy, but rather as a target biopsy [64]. When they cannot detect a tumor despite their efforts, image fusion can be recommended for biopsy.

Another issue can be raised because image registration is not perfect in fusing MRI and TRUS images. Lesions that are smaller or closer to the anterior capsule are frequently mis-targeted (Figure 13). Therefore, if an index lesion is small and anteriorly located, radiologists and urologists face a decreasing cancer detection rate when performing an MRI-TRUS image fusion biopsy [6].

We have experienced that almost all PI-RADS 4 or 5 lesions can be detected more precisely by using good TRUS protocols, imaging features, and biopsy techniques rather than with image fusion. Therefore, we can target a PI-RADS 4 or 5 lesion more precisely than we can with image fusion biopsy. Additionally, the number of biopsy cores can be reduced due to better tumor targeting (Figure 2 and Figure 7) [26]. However, systematic biopsy is difficult to skip during the image fusion biopsy procedures due to the relatively poor targeting of an index tumor [26]. Recently, a systematic review and meta-analysis reported that image fusion biopsy has a trend of improving cancer detection rate compared to cognitive biopsy [65]. However, they did not demonstrate that there is significant difference between image and cognitive fusion biopsies. Furthermore, the cognitive fusion biopsies they cited did not use the new TRUS protocols, imaging features, and techniques.

## 8. Conclusions

Radiologists and urologists who perform TRUS-guided biopsy in their practice can improve their tumor targeting if they become familiar with the new biopsy techniques and imaging features irrespective of the biopsy type (cognitive fusion or image fusion). Moreover, they should learn how to sample cores from the index tumor and when to add or skip a systematic biopsy.

## Figures and Tables

**Figure 1 cancers-13-05647-f001:**
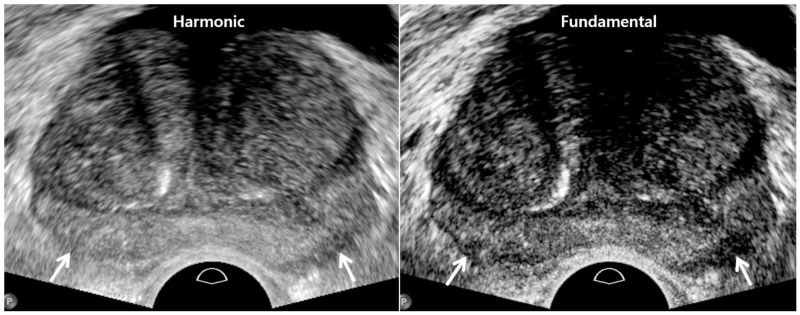
A 74-year-old man with high PSA (4.68 ng/mL): harmonic imaging (**left figure**) provides higher resolution compared to fundamental imaging (**right figure**). However, tissue contrast is clearer in fundamental imaging than harmonic imaging. As a result, posterior capsule (white arrows) is better depicted in fundamental imaging.

**Figure 2 cancers-13-05647-f002:**
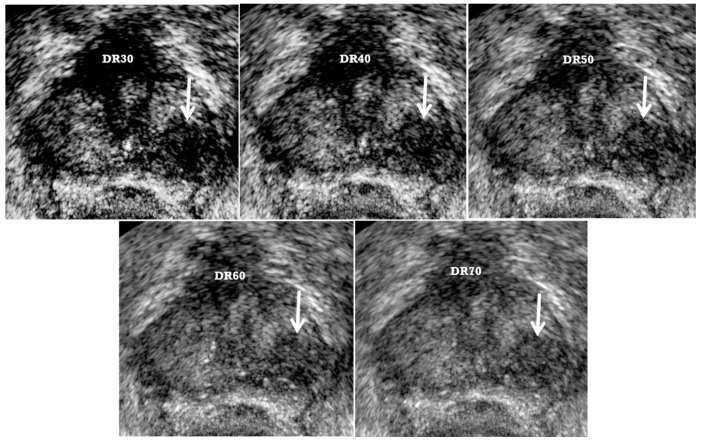
A 73-year-old man with high PSA (5.31 ng/mL): as dynamic range (DR) decreases from 70 to 30, a tumor (white arrow) becomes clear because of increasing tissue contrast. Subsequently, radiologists or urologists can detect and target it easily. The numbers of target and systematic cores were 4 and 4, respectively. Adnocarcinoma was confirmed with only target biopsy. The highest Gleason scores of tumor center and periphery were 8 (4 + 4) and 9 (4 + 5), respectively. Good lesion depiction could reduce the number of cores compared to 12-core systematic biopsy. However, as the DR decreases, gray-scale images becomes coarse because of decreasing tissue resolution. We recommend that the DR should be maintained from 40 to 50.

**Figure 3 cancers-13-05647-f003:**
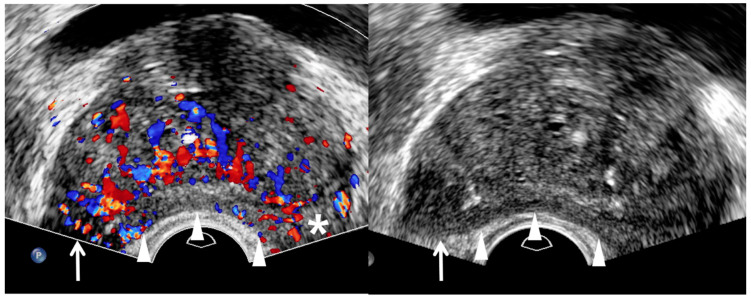
A 75-year-old man with high PSA (14.75 ng/mL): color Doppler TRUS image (**left figure**) shows a hypervascular area (white arrow) in the right mid-gland compared to the contra-lateral area (asterisk) in the left mid-gland. However, the right lesion is not clearly shown. Therefore, the tumor shape is completely depicted on gray-scale TRUS (**right figure**) when the tip of TRUS transducer is slightly moved to the right prostate. It is a hypoechoic tumor, which is projecting out from the capsule, suggesting extracapsular extension. White arrowheads indicate that the posterior capsule is compressed by a transrectal transducer, resulting in banana-shape deformity of the prostate. TRUS-guided target biopsy confirmed GS 8 (4 + 4) and post-biopsy MRI showed a T3a prostate cancer with right extra-capsular extension.

**Figure 4 cancers-13-05647-f004:**
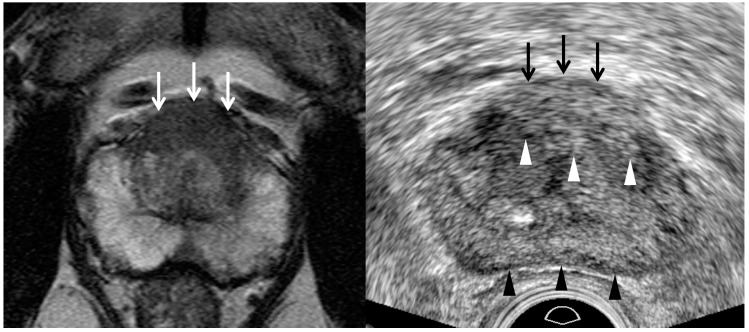
A 55-year-old man with high PSA (26.75 ng/mL): T2-weighted MR image (**left figure**) shows a PI-RADS 5 transition lesion (white arrows) in the anterior mid-line base. It is a lenticular homogeneous tumor in which the signal intensity is moderately hypointesne. Transrectal ultrasound (**right figure**) shows that it is a hyperechoic heterogeneous tumor. White arrowheads indicate the tumor margin which is irregular, infiltrative, and spiculate. Black arrowheads indicate the posterior capsule which is minimally compressed by a transducer. TRUS-guided target biopsy confirmed adenocarcinoma with Gleason score 8 (4 + 4).

**Figure 5 cancers-13-05647-f005:**
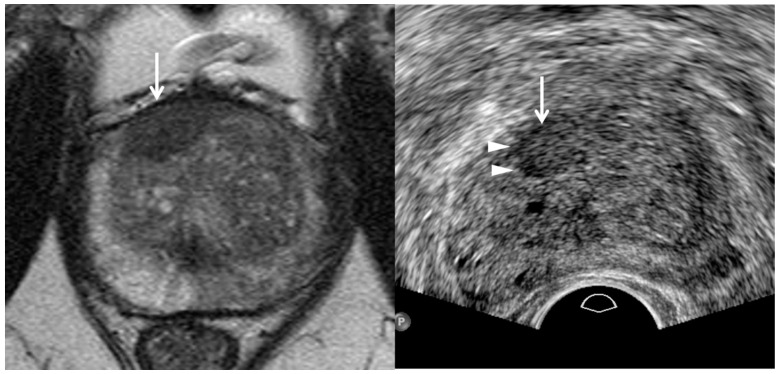
A 73-year-old man with high PSA (5.48 ng/mL): T2-weighted MR image (**left figure**) shows a 0.9 cm PI-RADS 4 transition lesion (white arrow) in the anterior right base. The tumor looks lenticular and homogeneous. In contrast, TRUS image (**right figure**) shows that it is detected in the anterior right mid-gland. It is slightly hyperechoic and a hypoechoic rim (white arrowheads) is detected at the right margin of the tumor. The histologic diagnosis was adenocarcinoma with GS 6 (3 + 3) with target biopsy.

**Figure 6 cancers-13-05647-f006:**
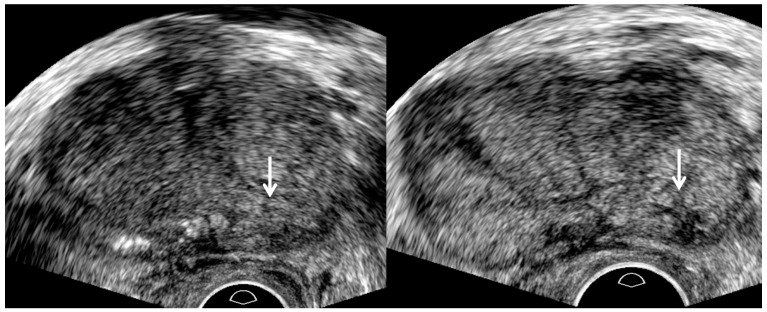
A 67-year-old man with high PSA (8.29 ng/mL): TRUS image (**left figure**) prior to biopsy shows an ill-defined lesion (white arrow) in the left base. Another TRUS image (**right figure**) after sampling two cores shows a well-defined lesion (white arrow) in the same location. The tumor margin becomes clear and spiculate because post-biopsy hemorrhage enhances tumor-to-normal tissue contrast. Accordingly, the relative length of adenocarcinoma with GS 7 (3 + 4) to normal tissue increases as the number of target cores increases.

**Figure 7 cancers-13-05647-f007:**
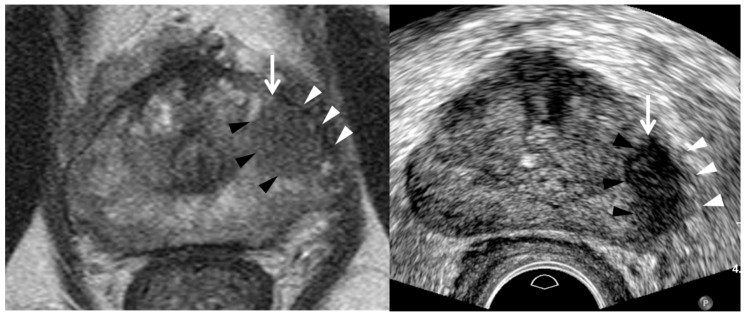
A 73-year-old man with high PSA (10.71 ng/mL): T2-weighted MR image (**left figure**) shows a PI-RADS 4 peripheral lesion (white arrow) in the left mid-gland. It is a 1.3 cm hypointense tumor in which the capsule (white arrowheads) is not disrupted, consistent with PI-RADS 4. Transrectal ultrasound image (**right figure**) shows it is a hypoechoic tumor (white arrow) with irregular or spiculate margin (black arrowheads). The capsule (white arrowheads) neighboring the tumor also appears thick and infiltrative, suggesting extra-capsular extension. Four target cores alone were obtained and systematic biopsy was skipped. The tumor was histologically confirmed adenocarcinoma GS 7 (4 + 3). The tumor shape looks more round on MRI than TRUS. Left peri-prostatic nerves were not saved because extensive capsular extension was detected during the robot-assisted radical prostatectomy.

**Figure 8 cancers-13-05647-f008:**
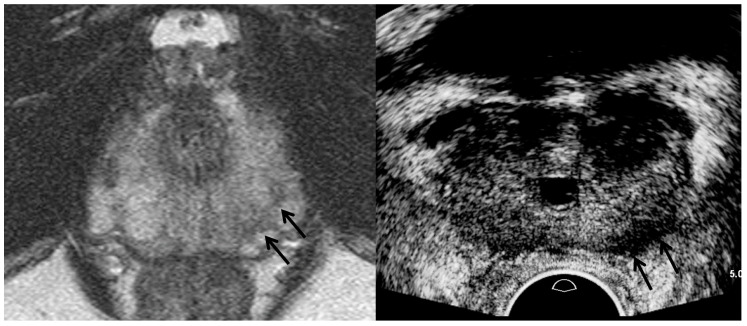
A 67-year-old man with high PSA (4.47 ng/mL): T2-weighted MR image (**left figure**) shows a PI-RADS 4 peripheral lesion (black arrows) in the left apex. However, TRUS image (**right figure**) shows it is a hypoechoic tumor (black arrows) in the left base. The tumor shape looks thicker on TRUS image compared to MR image. The tumor size on TRUS image is not identical on MR image, either. A great care should be taken to understand the difference between MRI and TRUS in terms of tumor location, shape, and size.

**Figure 9 cancers-13-05647-f009:**
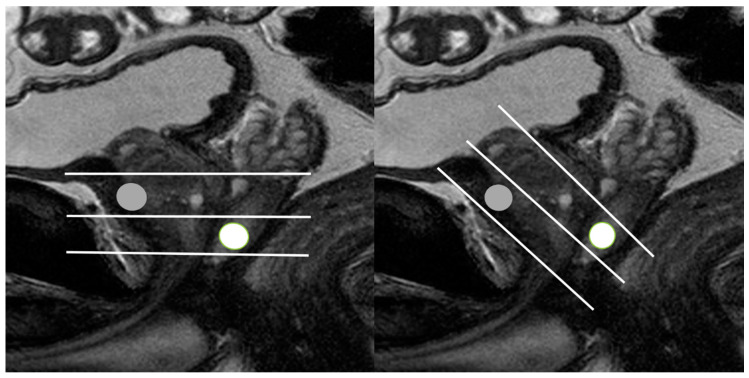
Schematic figures illustrating the different scan axes between MRI and TRUS: (**left figure**) and (**right figure**) figures indicate the scan axes (white lines) of MRI and TRUS, respectively. The MRI and TRUS scan axes are placed in the perpendicular and oblique direction to the prostate urethra, respectively. On MRI, an anterior lesion (gray circle) is located between mid-gland and base, while on TRUS, it is seen between mid-gland and apex. On MRI, a posterior lesion (white circle) is located between mid-gland and apex, while on TRUS, it (white circle) is seen between mid-gland and base.

**Figure 10 cancers-13-05647-f010:**
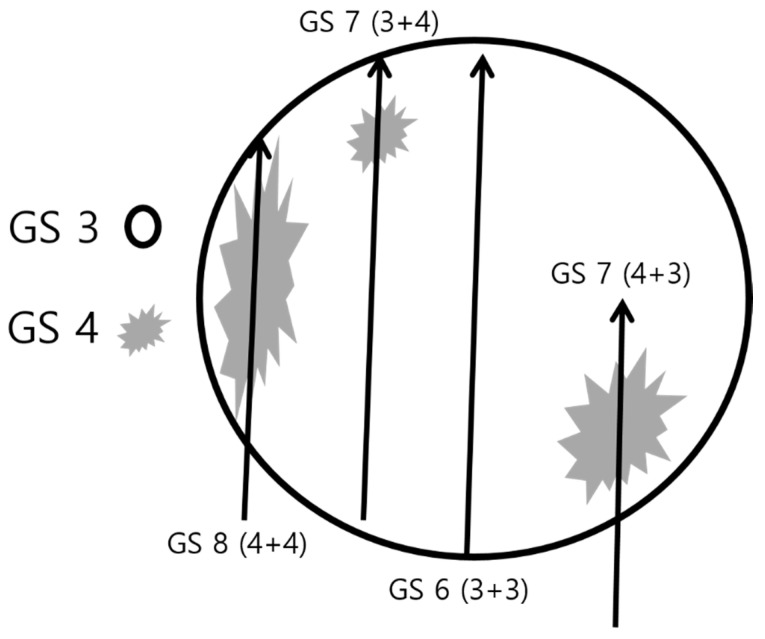
Schematic figure for multifocal sampling of an index lesion: a round circle indicates an index tumor in which Gleason score (GS) 3 (white area) and 4 (gray area) are major and minor tissues, respectively. If a radiologist or urologist samples cores only from the tumor center, the histologic diagnosis will be GS 6 (3 + 3), resulting in underestimation compared to that in prostatectomy. However, if cores are sampled from the tumor periphery, the histologic diagnosis will be GS 7 (3 + 4), GS 7 (4 + 3), or GS 8 (4 + 4). Therefore, multifocal sampling can reduce GS underestimation if biopsy cores are obtained from the peripheral areas as well as central areas of an index lesion.

**Figure 11 cancers-13-05647-f011:**
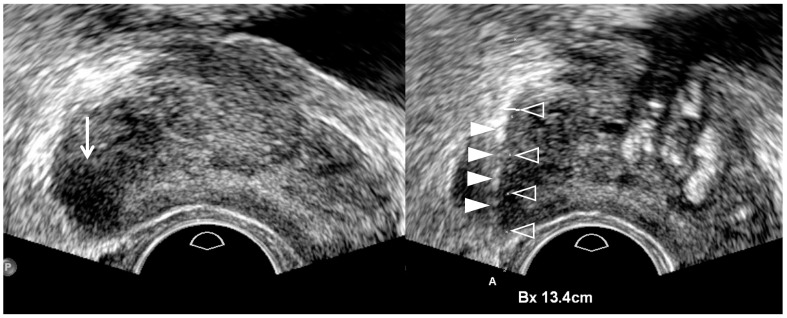
A 74-year-old man with high PSA (4.90 ng/mL): pre-biopsy TRUS image (**left figure**) shows a hypoechoic tumor (white arrow) in the right mid-gland. Post-biopsy TRUS image (**right figure**) is obtained when the tumor is targeted with a biopsy needle. Even though a needle guider is set on the probe and a biopsy guideline is displayed, the needle (solid arrowheads) is not completely aligned with the guideline (open arrowheads). Oligo-sampling may underestimate Gleason score compared to the prostatectomy.

**Figure 12 cancers-13-05647-f012:**
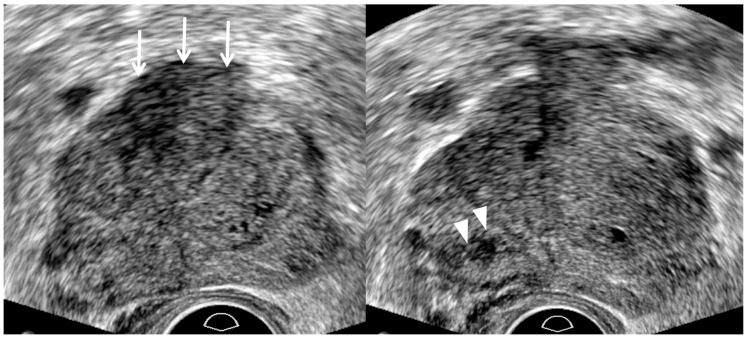
A 65-year-old man with high PSA (6.86 ng/mL): (**left figure**) TRUS image shows a 2 cm transition lesion (white arrows), which is categorized as PI-RADS 5 on T2-weighted MRI. (**Right figure**) TRUS image shows a 0.6 peripheral lesion (white arrowheads), which is categorized as PI-RADS 4 on diffusion-weighted MRI. Theoretically, because the PI-RADS 5 transition lesion is an index lesion, it is supposed to have a higher GS than the PI-RADS 4 peripheral lesion. However, the histologic diagnoses of the PI-RADS 4 and 5 lesions were GS 7 (3 + 4) and GS 6 (3 + 3) adenocarcinomas, respectively.

**Figure 13 cancers-13-05647-f013:**
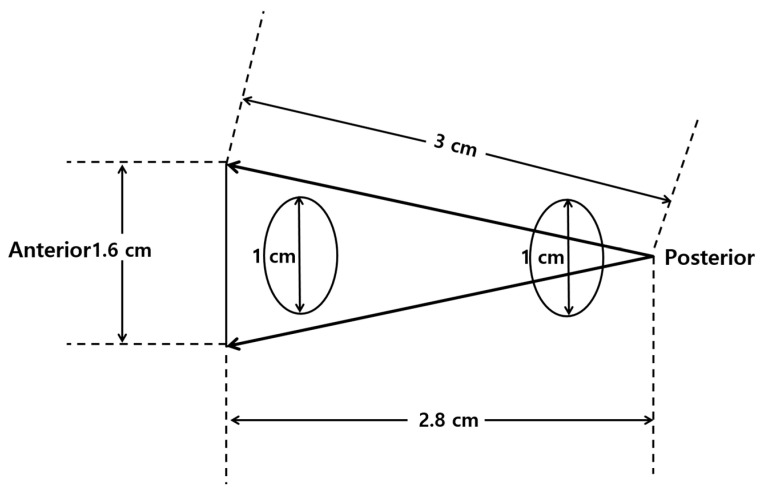
Schematic figure illustrating mis-targeting on image fusion biopsy: a schematic figure shows two oval lesions which are located near the anterior and posterior capsules. The posterior lesion can be targeted because it is close to the transducer. However, the same-sized lesion cannot be targeted because it is far from the transducer. The difference arises from the incomplete fusion of MRI and TRUS images. Mis-targeting increases as a tumor becomes small and far from the transducer.

**Table 1 cancers-13-05647-t001:** Comparison of old and new TRUS techniques for tumor detection.

TRUS Parameters	Old TRUS Techniques	New TRUS Techniques
US sequence	Harmonic imaging	Fundamental imaging
US artifacts	Rare	Frequent
Dynamic range	High	Low
Image resolution	High	Low
Tissue contrast	Low	High

TRUS, transrectal ultrasound.

**Table 2 cancers-13-05647-t002:** TRUS features of significant cancers.

TRUS Features	Peripheral Cancer		Transition Cancer	
	Insignificant Cancer	Significant Cancer	Insignificant Cancer	Significant Cancer
Echogenicity	Low	Lower	High	Higher
Size	Small	Large	Small	Large
Echotexture	Homogeneous	Heterogeneous	Homogeneous	Heterogeneous
Margin	Smooth	Irregular	Smooth	Irregular
Perfusion	Low	High	NA	NA
Hypoechoic rim	NA	NA	Clear	Unclear

TRUS, transrectal ultrasound; NA, not applicable.

**Table 3 cancers-13-05647-t003:** Different imaging features of tumors between MRI and TRUS.

Tumor Location/Morphology	MRI	TRUS
Scan axis to urethra	Perpendicular	Oblique
Anterior 1/3 location	Base and mid-gland	Mid-gland and apex
Middle 1/3 location	Same location	Same location
Posterior 1/3 location	Apex and mid-gland	Mid-gland and base
Tumor size	Different size	Different size
Tumor shape	Different shape	Different shape

MRI, magnetic resonance imaging; TRUS, transrectal ultrasound.

**Table 4 cancers-13-05647-t004:** Comparison of old and new TRUS biopsy techniques.

TRUS-Guided Biopsy	Old Biopsy Techniques	New Biopsy Techniques
Central tumor targeting	Yes	Yes
Peripheral tumor targeting	No	Yes
Number of target cores	Oligo-cores (1–3)	Multi-cores (4–6)
Systematic biopsy	Yes	Yes, but no in T3/T4 cancers

TRUS, transrectal ultrasound.

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
