# Peer review of "How to Improve TRUS-Guided Target Biopsy following Prostate MRI"

_cancers, 2021, doi:10.3390/cancers13225647_

Round 1
Reviewer 1 Report
It is a comprehensive and practical review paper. Thanks for invitation.
The authors reported “ How to Improve TRUS-guided Target Biopsy Following Prostate MRI” and claimed a new biopsy techniques including how to understand different tumor location, size, and shape between prostate MRI and TRUS and how to target an index lesion regarding biopsy strategy and cores. This report is interesting and reasonable. Clinical physicians who perform TRUS-guided biopsy in their practice can improve their tumor targeting if they become familiar with the new biopsy techniques and imaging features(fundamental imaging and harmonic imaging) irrespective of the biopsy type (cognitive fusion or image fusion). Then, they could learn how to sample cores from the index tumor and add or skip a systematic biopsy.
Author Response
Authors responses to Reviewer #1 comments
It is a comprehensive and practical review paper. Thanks for invitation.
The authors reported “How to Improve TRUS-guided Target Biopsy Following Prostate MRI” and claimed a new biopsy techniques including how to understand different tumor location, size, and shape between prostate MRI and TRUS and how to target an index lesion regarding biopsy strategy and cores. This report is interesting and reasonable. Clinical physicians who perform TRUS-guided biopsy in their practice can improve their tumor targeting if they become familiar with the new biopsy techniques and imaging features (fundamental imaging and harmonic imaging) irrespective of the biopsy type (cognitive fusion or image fusion). Then, they could learn how to sample cores from the index tumor and add or skip a systematic biopsy.
Response: Thank you for your comments. If radiologists or urologists are familiar with new TRUS protocols, new TRUS features of peripheral or transition cancer, and imaging difference between MRI and TRUS, they will improve targeting an index lesion. Besides, if they can learn how to sample cores from an index lesion and when to add or skip a systematic biopsy, these biopsy procedures may contribute to reducing GS underestimation without increasing complication rate.
Reviewer 2 Report
This study was reported the utility of TRUS-guided biopsy for detecting prostate cancer. Overall, this paper is well written. The reviewer would like to suggest some critiques as follows.
Major revision
- On line 47, the authors described that TRUS image is good quality and essential. The reviewer thinks that the reproducibility of US image is not sufficient to detect prostate cancer than TRUS. The authors should explain this point.
- On line 48, “which is detected on MRI” is unclear? Prostate cancer is detected ?
- On the section 6, the authors describe that systematic biopsy is not necessary for detecting prostate cancer. However, target biopsy has vital weakness. With regard to PI-RADS 3 lesions, the csPCa detection rate of target biopsy alone was found to be significantly lower than that of combined target and systematic biopsy. The reviewer thinks that combined biopsy may also be necessary for the detection of clinically significant prostate cancer. The authors should describe this point more detail.
- Today, the reviewer thinks that many urologist do not perform cognitive fusion biopsy. Therefore, the authors may delete this section.
Author Response
Authors responses to Reviewer #2 comments
This study was reported the utility of TRUS-guided biopsy for detecting prostate cancer. Overall, this paper is well written. The reviewer would like to suggest some critiques as follows.
Major revision
1. On line 47, the authors described that TRUS image is good quality and essential. The reviewer thinks that the reproducibility of US image is not sufficient to detect prostate cancer than TRUS. The authors should explain this point.
Response: Thank you for your comment. The quality of TRUS image is of great importance to find an index lesion which is detected on MRI. It is influenced mainly by two key protocols such as using fundamental imaging and low dynamic range to increase tissue contrast between index tumor and normal tissue. If radiologists or urologists become familiar with these TRUS protocols, they can produce reproducible TRUS images in which an index lesion is identified. These statements will be added.
2. On line 48, “which is detected on MRI” is unclear? Prostate cancer is detected?
Response: Thank you for your comment. It is an index tumor. I will rephrase the statement for readers to understand better.
3. On the section 6, the authors describe that systematic biopsy is not necessary for detecting prostate cancer. However, target biopsy has vital weakness. With regard to PI-RADS 3 lesions, the csPCa detection rate of target biopsy alone was found to be significantly lower than that of combined target and systematic biopsy. The reviewer thinks that combined biopsy may also be necessary for the detection of clinically significant prostate cancer. The authors should describe this point more detail.
Response: Thank you for your comment. Definitively, I agree with you. Systematic biopsy becomes important in detecting a significant cancer in PI-RADS 1-3 lesions because target biopsy alone does not yield so high significant cancer detection rate. I will add this statement.
4. Today, the reviewer thinks that many urologists do not perform cognitive fusion biopsy. Therefore, the authors may delete this section.
Response: Thank you for your comment. I understand what you mean. However, I do not mean to say that urologists perform image fusion biopsy alone. If urologists as well as radiologists are familiar with new TRUS protocols, new TRUS findings, and new biopsy techniques, they can target an index tumor precisely regardless of the biopsy types (cognitive fusion vs image fusion). However, many radiologists and urologists do not know difference between cognitive and image fusion biopsy in terms of tumor targeting. Therefore, section 7 just helps them to understand it.
Round 2
Reviewer 2 Report
The authors revised this study according to the reviewer’s recommendation. The reviewer believes that this paper will provide useful information for the readers.